# Does PM2.5 (Pollutant) Reduce Firms’ Innovation Output?

**DOI:** 10.3390/ijerph20021112

**Published:** 2023-01-08

**Authors:** Zhiqiao Xiong, Dandan Li, Hongwei Yu

**Affiliations:** 1School of Economics and Management, Changsha University of Science and Technology, Changsha 410076, China; 2School of Low Carbon Economics, Hubei University of Economics, Wuhan 430205, China; 3Collaborative Innovation Center for Emissions Trading System Co-Constructed by the Province and Ministry, Hubei University of Economics, Wuhan 430205, China; 4Institute of Quality Development Strategy, Wuhan University, Wuhan 430072, China

**Keywords:** PM2.5, innovation output, instrument variable, thermal inversions, L25, Q51, Q53

## Abstract

The potentially serious economic consequences of China’s severe air pollution problem cannot be overlooked, especially the impact on corporate innovation, which is a core driver guiding firms towards efficient and high-quality development. This paper explores the direct effect and mechanism of PM2.5 on firms’ innovation output through the identification strategy of instrument variable. Based on the data of Listed Companies in China from 2003 to 2016, we used thermal inversion as the instrument variable for PM2.5 for estimation. The results show that each 1 ug/m^3^ increase in PM2.5 concentration causes an average reduction in innovation output of listed companies by about 7.0%. The test of “Porter hypothesis” shows that environmental regulation has not encouraged firms to innovate more. We further used the 2013 China Social Survey (CSS) data to verify the human capital mechanism of PM2.5 affecting firm innovation at micro level. The results show that PM 2.5 deteriorates the healthy human capital in a firm, which reduces the innovation output. This article helps to understand the relationship between air pollution and firms’ innovation and to develop appropriate policies.

## 1. Introduction

Innovation is an important engine for enterprises to achieve long-term development and is an inexhaustible driving force for economic growth. However, China’s air pollution problem, especially PM2.5 pollution, is serious. More than half of China’s cities exceeded air pollution concentrations, and the number of days with PM2.5 as the primary pollutant accounted for 78.8% of the days with severe and heavy pollution. It is estimated that the annual number of deaths attributable to PM2.5 pollution in China has increased to 971,000 by 2017 In the context of such severe pollution in China [1], the burden on firms to carry out innovative activities continues to increase, so how will PM2.5 affect firms’ innovation? What are the mechanisms at play? These questions need to be answered.

In this paper, thermal inversions are used as the instrumental variable of PM2.5 to analyze the effect and mechanism of PM2.5 on micro-firm innovation. In terms of impact effect, we take the innovation output of Chinese listed companies in 2003–2016 as the analysis object. The results of IV estimation model show that PM2.5 leads to the reduction in firm innovation output. Specifically, for every 1 ug/m^3^ increase in the annual average PM2.5 concentration in the region, the total number of patent applications and the total number of invention patent applications of listed companies decreases by about 7.0% on average. At the same time, we further control the environmental protection policies and the number of regional environmental regulations, as well as other environmental factors such as pollutant emissions. The results show that PM2.5 does not have a significant impact on the innovation output of individual firms through environmental regulation, and the “Porter hypothesis” of PM2.5 and firm innovation does not hold.

In terms of the mechanism of action, we used the 2013 Chinese Social Survey (CSS) data to verify the human capital mechanism from the individual micro level. IV estimates indicate that regional PM2.5 can significantly worsen the health of individual laborers, leading to a decline in labor productivity, lower job satisfaction, and subjective well-being, and increase their negative emotions at work. In general, PM2.5 significantly worsens the human capital of firms, which may lead to the reduction in innovation output of firms. This verifies that the mechanism of the human capital of PM2.5 affects firm innovation. At the same time, we show that there is an inverted U-shaped relationship between PM2.5 and firm innovation output, and analyze the heterogeneity of PM2.5 on firm innovation output from the perspectives of PM2.5 pollution levels, capital intensity, and labor productivity. The results also confirm the existence of the human capital mechanism, and the environmental regulation mechanism is not significant.

This paper has the following contributions: First of all, this paper expands the research on the influence of PM2.5 on micro-firms to the field of firm innovation. Studies on air pollution on inventory level [2], and market value have been carried out [3], but research on firm innovation output is rarely seen. Secondly, this paper further explores the external environmental factors that affect firm innovation, and further enriches the relevant research on the factors influencing firm innovation. Existing research on the factors affecting the innovation output of firms mainly focuses on the internal factors of the firm, such as salary incentives and manager characteristics [4,5]; or external market factors, such as market competition [6]; or macro policy factors such as government subsidies [7]; but specifically the air pollution phenomenon of PM2.5 has not been seen yet. Thirdly, this paper finds that environmental regulation and other policies do not affect the role of PM2.5 in firm innovation, that is, PM2.5 does not have a significant impact on firm innovation through policies such as environmental regulation. Whether environmental regulation has an impact on firm innovation has been controversial in the academic world. The more famous assertion is the “Porter hypothesis”, which considers that environmental regulation has a negative impact on firm production innovation in the short term, but promotes technological innovation in the long run [8]. However, the research in this paper shows that the “Porter hypothesis” has not been verified in the air environment. Finally, the paper further validates the healthy human capital mechanism of PM2.5 affecting firm innovation.

The rest of this paper is organized as follows: Section 2 is literature Review; Section 3 is about the setting of the model, the measurement of variables, and the source of data; Section 4, the estimation the effect of PM2.5 on the innovation output of firms; Section 5, the test the mechanism of action; and Section 6 is the summary and discussion.

## 2. Literature Review

Air pollution has a wide-ranging impact on various social and economic actors. Since the effect of air pollution on individual residents is considered relatively direct and easy to observe [9,10], a large number of literatures have been extensively studied [11,12,13,14,15,16,17]. The overall results show that pollutants such as PM2.5 can worsen the physical and mental health of individual residents and lead to the loss of social welfare [18,19,20], lower birth rate [21], increased mortality [22,23], increased traffic fatalities [24], increased crime rate [25], population mobility [26,27], and many other social issues. As the core subject of economic activities, enterprises are also affected by external air pollution. Comparatively, the impact of air pollution on firms is not as easy to observe as individual residents, so research on the impact of air pollution on firms is rare.

The early research on the impact of air pollution on firm subject was mainly indirectly identified through the explicit variable of “environmental regulation”, and the research conclusions did not obtain a more consistent consensus. Some scholars believe that the increase in air pollution is mainly caused by the expansion of industrial production scale or renewable energy consumption scales [28], and large-scale industrial production can bring higher production efficiency [29,30]; on the contrary, if air pollution emissions are controlled, the production cost of the firm is increased and the productivity is reduced [31,32]. This can be verified in empirical studies in the United States, Europe, and China [33,34]. Another group of scholars believe that although local governments can strengthen environmental regulation [35], which may impose a burden on the production cost of firms in the short term, in the long run it will stimulate firms to carry out technological innovation or process innovation, and ultimately improve the production efficiency [8,36,37]. These studies did not directly address the impact of air pollution on firms’ productivity, but attributed the change in productivity caused by reduced air pollution to the result of environmental regulation, and cannot obtain an accurate estimate of the impact of air pollution on firms’ productivity.

In recent years, some scholars have tried to estimate the direct impact of air pollution on firms by designing and constructing some sophisticated identification strategies. For example, some scholars used the closure of a large oil refinery as a quasi-natural experiment to study the impact of air pollution on the labor supply of firms [38]. The results show that the reduction in air pollution will improve the living and production environment of workers, increase labor supply, and improve the efficiency of workers; Some scholars constructed a quasi-natural experiment based on the traffic dependence of the industry, and used methods such as instrumental variables to verify that air pollution significantly increases the inventory level of firms, and machine learning can alleviate inventory backlogs by predicting product demand [39]. These studies provide good ideas and methods for further analysis of the impact of air pollution on the micro-firms.

Although scholars have begun to study the impact of air pollution on the labor supply, productivity, and inventory of micro-firms through the design of more accurate identification strategies, the research on the theme of firm innovation needs further development. Existing domestic and international literature on environmental pollution and firm innovation is carried out around environmental regulation or the “Porter hypothesis” [8,40,41], and does not explore the direct impact of PM2.5 on individual firm innovation, and high concentration of PM2.5 is one of the important reasons for the formation of hazy weather [42]. Similar to the foregoing, existing research of this type may have the following disadvantages: The first is to attribute the role of pollution to firm innovation to “environmental regulation”, which may lead to bias in the estimation of pollution effects [43,44]; the second is that the literature has different measures and standards for “environmental regulation” [45,46], this recognition error may lead to inconsistencies in research findings [47]. For example, for research in China, some scholars have verified the existence of the “Porter hypothesis” [48,49,50], but other scholars’ empirical research does not support the argument of the Porter hypothesis [51,52]; finally there is the possibility of missing important influence mechanisms, such as the effects of pollution on individual health, labor productivity, and mobility [26,38,53]. Since human capital is the first driver of innovation, while air pollution can seriously affect the health of human capital [54], it is reasonable to suspect that human capital is an important mechanism by which PM2.5 affects corporate innovation.

In view of the above problems, this paper focuses on the direct impact and mechanism of PM2.5 on micro-individual firm innovation. Based on the existing literature, this paper focuses on the use of instrumental variables to build a more accurate identification strategy. PM2.5 is related to regional economic and social activities on the one hand, and local natural climatic conditions on the other. It can generally be considered that the long-term stable natural climatic conditions in a region do not have an impact on regional economic or social activities. Therefore, if we can find a natural climate variable that directly affects the formation of PM2.5 in the region as an instrumental variable, the effect of PM2.5 on the innovation of micro-individual firms is more accurately identified. Thermal Inversions may be such a naturally occurring meteorological phenomenon. Under normal circumstances, the temperature in the troposphere decreases with the increase in altitude. This kind of atmospheric stratification is prone to convective movement, and the pollutants in the near-surface layer can be dissipated to high altitude or even far away, thus reducing the degree of urban air pollution. However, under certain weather conditions, the atmospheric structure experiences an abnormal phenomenon where the temperature increases with height, resulting in stable atmosphere. The occurrence of inversion temperature is not conducive to the rising movement of air, so that low-level water vapor and pollutants cannot be diffused out to high altitude, which causes fog formation and accumulation of pollutants. The occurrence of inversion directly leads to an increase in air pollution, but at the same time, inversion is a naturally occurring meteorological phenomenon, its formation not affected by regional economic or social factors, so many economists use it as an instrumental variable for regional air pollution [55,56,57].

## 3. Models, Variables and Data

### 3.1. Model Setting and Measurement Strategy

In this paper, the firm’s innovation output is the micro-firm level data, the PM2.5 is the city-level data, and the main control variables are the relevant factors at the firm level. For factors outside the firm level, a fixed-effects model is used for processing, including the control of regional and temporal fixed effects. This paper sets the following quantitative analysis model:(1)$$I_{\mathrm{ijt}}=\alpha_{0}+\alpha_{1}P_{jt}+\alpha_{2}X_{ijt}+\delta_{j}+\rho_{t}+\varepsilon_{ijt}$$

The explanatory variable $I_{\mathrm{ijt}}$ in Equation (1) is the patent case of firm i in the t year of city j; the core explanatory variable $P_{jt}$ is the annual average of PM2.5 in the t year of city j; $X_{ijt}$ is the control variable, According to the relevant literature, it mainly includes other general factors that may affect the innovation output of firms within the firm, such as age, scale, investment, capital structure, etc. $\delta_{j}$ is the fixed effect of the regional city, $\rho_{t}$ is the time fixed effect, and $\varepsilon_{ijt}$ is the error term.

The main goal of this paper is to analyze the direct effect of PM2.5 on firm innovation. The core strategy chooses inversion as the instrumental variable of regional PM2.5 and uses two-stage least squares (2SLS) for regression analysis. At the same time, in order to examine the impact of environmental regulation on firm innovation, in the regression of instrumental variables, group regression is further carried out according to the implementation of regional environmental regulation and innovation policy to test the original estimation results.

On the basis of examining the impact of environmental regulation on firm innovation, this paper focuses on the possible mechanism of PM2.5 affecting firm innovation from the perspective of human capital. This paper chooses to use the micro survey data of individual residents to test the impact of PM2.5 on labor health human capital and labor loss, and construct a quantitative analysis model as shown below:(2)$$M_{\mathrm{ij}}=\beta_{0}+\beta_{1}P_{j}+\beta_{2}Z_{ij}+\delta_{j}+\varepsilon \mu_{ij}$$

Equation (2) is an analytical model at the individual level. Among them, the explanatory variable $M_{\mathrm{ij}}$ is the relevant situation of the human capital of the ith investigated labor in the city j; the core explanatory variable $P_{j}$ is the annual average of PM2.5 of the city j.

### 3.2. Variable Selection and Data Source

#### 3.2.1. Firm Innovation

This paper selects firm patent data as a proxy indicator for firm innovation. There are many variables that reflect firm innovation, such as R&D expenditures and personnel inputs that reflect innovation process inputs [58,59], and variables such as the number of patents and new products that reflect the output of the innovation results [60,61]. In contrast, this paper observes the final impact of PM2.5 on firm innovation from the perspective of outcome output, and therefore does not consider innovation input variables; for the resulting variables, the number of patents is easier to observe directly and objectively than the indicators such as new products. In terms of data sources, this paper selects the data of Chinese listed companies in the period from 2003 to 2016 for analysis.

#### 3.2.2. PM2.5

There are many sources of data for PM2.5, such as environmental monitoring stations [27,62,63], spatial remote sensing information [64,65], etc., the choice of PM2.5 data sources can be considered from the spatial range and time frequency. In the spatial dimension, PM2.5 is usually a regional value, and more regional values are used at the city level. In comparison, the geographical distribution of environmental monitoring stations at the urban level is not uniform, and there are even no environmental monitoring stations in some areas with low economic development [66], while the area covered by spatial remote sensing information is more extensive and homogeneous. In the time dimension, PM2.5 is usually a value, in which the data of the environmental monitoring station is more real-time dynamic, while the spatial remote sensing information has a relatively long time interval [67].

It can be said that the various PM2.5 data sources have their own advantages. The object of this paper is that firm innovation is a micro-data at the individual firm level. Therefore, this paper chooses spatial remote sensing information that is more advantageous in space as the source of PM2.5 data in this paper. In terms of time, this paper considers the time frequency of the listed company’s annual report data, and selects the annual urban PM2.5 average concentration, which also avoids the problem of the spatial remote sensing information monitoring time interval. Specifically, this paper selects PM2.5 data from the Center for International Earth Science Information Network (CIESIN)/Columbia University, which is hosted by the National Aeronautics and Space Administration (NASA). CIESIN developed and produced a global PM2.5 continuous monitoring algorithm system by using space remote sensing satellites and multi-angle imaging spectrometers to scientifically solve the problem of converting spatial remote sensing information into PM2.5. They provide PM2.5 concentration surface data with a spatial resolution of 0.5° × 0.625° (equivalent to 50 square kilometers) with a grid length ranging from 70 degrees north latitude to 60 degrees south latitude. On this basis, this paper further uses ArcGIS software to extract Chinese data from global datasets and uses the method of Inverse Distance Weighting (IDW) [68,69,70], matching it with the latitude and longitude coordinate system of China’s prefecture-level cities, and obtaining PM2.5 concentration data of prefecture-level cities in China.

#### 3.2.3. Thermal Inversion

In order to better match the PM2.5 data, this paper also obtains the Thermal Inversions data from the space remote sensing information released by the National Aeronautics and Space Administration (NASA). NASA divides the Earth into a grid with a latitude and longitude interval of 0.5° × 0.625°, and from 1980 onwards has continuously reported the temperature of 42 different sea levels every six hours. In this paper, we obtain the three sea level temperature data closest to the ground. First, we average the sea level temperature of each grid every day and count the first layer temperature lower than the second layer as one inversion temperature, then accumulate the total inversion days of each natural year of each grid. Finally, we determine the number and weight of grids covered by each prefecture-level city in China according to the IDW method, and finally calculate the annual number of inversion days in each prefecture-level city. In order to test the robustness, this paper also calculates the case where the first layer temperature is lower than the third layer.

#### 3.2.4. Other Variables and Data

The goal of this paper is to accurately estimate the direct impact of PM2.5 on the firms’ innovation output. On the one hand, it is necessary to eliminate the influence of environmental regulation on firm innovation as much as possible. On the other hand, it is necessary to explore the direct mechanism of PM2.5 affecting the innovation output of firms.

In terms of eliminating the impact of environmental regulations, this paper uses policy changes to reflect the impact of environmental regulations. Although existing descriptions of environmental regulations are abundant, such as pollutant discharge [71], energy consumption density [72], pollution abatement costs [73], environmental governance costs [46], environmental regulation is usually directly observable through regional policy changes [74,75]. In terms of the direct action mechanism, this paper focuses on the mechanism by which PM2.5 acts on firm innovation through human capital. The firm data in the database of Chinese listed companies does not contain relevant variables sufficient to reflect individual human capital. This paper chooses to use the 2013 Chinese Social Survey (CSS) data from the Chinese Academy of Social Sciences to test the micro-mechanism.

The data description statistics of each research variable in this paper are shown in Table 1.

## 4. Results and Analysis

### 4.1. Regression Result

In the regression, we focus on the innovation of patents and examine the impact of regional PM2.5 on firm innovation from two aspects. First, we examine the impact of PM2.5 on the number of innovative firms, including the total number of patent applications applied for and the total number of invention patent applications filed. Second, we further investigate the impact of PM2.5 on the total number of other types of patent applications, including utility model patents and design patents. The regression is analyzed according to Equation (1) using a two-way fixed-effect model and an instrumental variable model.

Table 2 shows the results of the impact of PM2.5 on the number of patent applications filed. The results of fixed-effect regression estimation show that the impact of regional PM2.5 concentration on the total number of patent applications and the total number of invention patent applications is not significant; and the estimation results of the instrumental variables show that the PM2.5 has a significant negative effect on the number of patent applications of firms. Specifically, for every 1 ug/m^3^ increase in the annual average PM2.5 concentration in the region, the total number of patent applications and the total number of invention patent applications of listed firms falls by about 7% on average, that is, the PM2.5 significantly reduces the firm’s innovation output, and the “Porter hypothesis” is not established. The validity of the instrumental variables passed the tests. Similarly, the analysis of patent application data also confirms that there are serious endogenous problems between PM2.5 and firm innovation. The estimation of instrumental variables in this paper may be more accurate.

In Table 3, we further use the instrumental variable method to examine the effects of PM2.5 on other types of patent applications and actual authorizations. The results show that PM2.5 has a significant negative impact on the number of applications and the amount of licensing for utility models and design patents. On average, for every 1 ug/m^3^ increase in the annual PM2.5 concentration in the region, the number of utility model patents and grants of listed firms is reduced by about 6%, and the number of applications and licenses for design patents decreases by approximately 2.6% and 3.7%, respectively. Compared with the estimation results in Table 3, the negative impact of PM2.5 on the number of invention patent applications (6.9%) is greater than that of utility models (6.3%) and design (2.6%). It is generally believed that invention patents are more innovative than utility models and design-type patents [76], and are more representative of the original level of innovation of a firm [77]. Therefore, the effect of PM2.5 on the original innovation of firms is more significant.

### 4.2. Robustness Test

This paper further uses the instrumental variable method to test the multi-faceted robustness of the baseline estimates. First, to control the influence of regional and industry-level factors that may change over time, we examined the fixed-effect control of regions, time, and industry. Second, to control for selection bias due to different sample sizes, we examined the estimates of the sample sizes of different firms. Third, to control for the possible effect of cities heterogeneity on the estimation results, we examined the estimates of the characteristics of different cities. Fourth, to control for bias caused by the explanatory variable measures, we replaced the dependent variable measures. Finally, we replaced the ordinary robust standard errors with clustering robust standard errors.

Table 4 shows the fixed effect estimates that control different regions, times, industries, and their interactions. The results show that after controlling more fixed effects, although the absolute value of most coefficients is reduced compared with the baseline regression (about 5.5%), it is basically kept at the 1% level, which verifies the conclusion of the baseline regression.

Table 5 shows the estimated results considering the sample of different listed firms. First, we excluded sample firms that did not apply for patents, the results showed that PM2.5 led to a decline in the innovation output of these firms. Although the significance of the coefficient and the absolute value of the coefficient are lower than the baseline regression, it can remain significant at the 5% level. Then we focused on the sample of manufacturing firms, and the estimated results are not much different from the baseline regression results. Finally, we only observed the sample firms that survived between 2003 and 2016 in the listed firms, and analyzed the balance panel, the absolute value of the coefficient is more than doubled compared with the baseline regression. The reliability of the baseline results is also validated by considering the estimation results of different firm samples.

This paper further examines the robustness from the regional urban level, the results are shown in Table 6. First, we removed the sample firms from the provincial and sub-provincial cities, the results showed that there was no significant change in the coefficient significance compared with the basic regression, and the absolute value of the coefficient increased, indicating that the PM2.5 has a greater negative impact on the innovation of listed firms in the prefecture-level cities. Then we control the series of urban characteristic variables, including GDP per capita, unemployment rate, human capital, education expenditure, population density, government science and technology expenditure, etc., and the estimation results are not much different from the baseline regression results. Urban city level inspections further validate the reliability of baseline regression.

The dependent variable in the baseline regression is the total number of patent applications, and this paper further replaces the dependent variable with the share of patents for robustness testing, and the results are shown in Table 7. From the results, it can be seen that the regression coefficient is significantly negative, which again indicates the robustness of the findings of this paper.

Ordinary standard errors are used in the benchmark regression, and this paper further uses clustered robust standard errors for robustness testing, and the results are shown in Table 8. From the results, it is clear that the results are consistent with the results of the benchmark regression, whether the standard errors are clustered to city, province, or industry–year, city–year, or city–industry.

### 4.3. Test the Effects of Environmental Regulations

Many extant literatures suggest that environmental regulation has an impact on firm innovation [8,47], and the goal of this paper is to estimate the direct effect of PM2.5 on firm innovation; therefore, it is necessary to control or eliminate the impact of environmental regulations. Quantitative measurement of environmental regulations is not easy, the literature has described using a number of different variables, such as firm pollution investments [78], pollutant emissions [71], and pollution abatement costs [73]. Such a description directly internalizes environmental regulation into the specific behavior of the firm, which may lead to bias in the estimation results.

This paper argues that environmental regulation itself is mainly embodied in the form of external policies. This paper sorts out relevant policies related to environmental regulation in the country and cities, including the “Ecological Demonstration Area” (China awarded the title of recognition to units with outstanding work achievements during the construction of ecological demonstration areas, and set up seven batches of recognition districts and counties), “Atmospheric Ten Articles”(The “Ten Atmospheric Measures” policy issued by China, prepared by the Ministry of Environmental Protection in collaboration with relevant departments, proposes 10 articles and 35 specific measures to address the problem of air pollution), and “Two Control Zones”(China has produced, may produce acid rain or other areas of serious sulfur dioxide pollution, designated as acid rain control area or sulfur dioxide pollution control area). Since these environmental policies are based on the scope of the city, in the specific estimation, we put the interaction items of these policy dummy variables and urban fixed effects into the baseline model for control. At the same time, this paper also collects the number of laws and regulations related to environmental protection in each city, and adds it directly as a control variable to the IV estimation model. In addition, we also consider the impact of other pollutants in various cities (such as SO_2_ and wastewater). We control the impact of environmental regulations by adding pollutant emissions to the IV baseline regression model.

Table 9 shows the estimated results of the impact of environmental regulations. Regardless of the total number of patent applications or the number of invention patents, after controlling the variables of environmental regulations in each region, compared with the baseline IV regression results, the IV estimation results are significant at the 1% level, and the coefficient size is also basically maintained at a value of about 7%, the overall difference is not large. This shows that environmental regulation does not significantly affect the role of PM2.5 in firm innovation, or that PM2.5 does not have a significant impact on individual firm innovation output through environmental regulation. Therefore, the role of the environmental regulation mechanism that PM2.5 affects firm innovation is not obvious, and the “Porter hypothesis” of PM2.5 and firm innovation is not established.

## 5. Mechanism Testing and Heterogeneity Analysis

### 5.1. Human Capital Mechanism Test

The effect of PM2.5 remains after controlling for regulation, then we need to find other possible mechanisms to further explore the significant channels of PM2.5 effect on individual firm innovation output. A large number of studies on the effects of air pollution on individuals indicate that air pollution has many negative effects on individual labor, such as physical and mental health problems [53,79,80,81], leading to a decline in individual labor productivity [38,82], labor migration and loss [83], and so on. Based on this, we suspect that the impact of PM2.5 on individual firm innovation output is likely to be generated by human capital mechanisms. In this section we use CSS data to test the mechanism. According to the relevant literature, we consider the analysis of individual labor health, labor productivity, subjective satisfaction and happiness, and work psychology.

We use the instrumental variable model to estimate, and the results are shown in Table 10. The results in columns (1) and (2) show that PM2.5 significantly increases the medical expenditure of the individual labor force and significantly reduces their health satisfaction; the results in column (3) verify that PM2.5 significantly reduces the productivity of workers; the results in columns (4) and (5) indicate that PM2.5 significantly reduces the worker’s job satisfaction and happiness; columns (6) to (9) show that PM2.5 can significantly reduce the positive emotions of workers at work (such as pleasure and enjoyment, etc.), while also significantly increase negative emotions (such as anger, disgust, etc.). The results in Table 9 are a good example of our hypothesis that PM2.5 can significantly worsen a firm’s human capital, which in turn may lead to a reduction in firm innovation output. Therefore, the human capital mechanism that PM2.5 affects firm innovation may exist.

### 5.2. Further Testing of the Mechanism

If the effect of PM2.5 on the innovation output of micro-individual firms is through the human capital mechanism, then we have reason to speculate that when the degree of PM2.5 pollution is low, PM2.5 has no negative impact on firm innovation output. This is because when the PM2.5 pollution is below a certain level, the PM2.5 no longer has a significant impact on an individual’s physiology or psychology, and of course it does not affect the firm’s innovation output through the human capital of the firm.

We further examine the possible effects and mechanisms of PM2.5 on the level of innovation output in the lower pollution levels. When we analyzed the low pollution level, the human capital mechanism that PM2.5 affects firm innovation does not exist. At the same time, we also believe that when the PM2.5 is at a low pollution level, the environmental regulation mechanism that affects the innovation of the firm does not appear, because many scholars have noted that the necessity of environmental regulation is weakened or even eliminated when the level of pollution is low [46]. We suspect that when PM2.5 is in the low pollution range, it may have a positive impact on firm innovation. From the mechanism analysis, the increase in PM2.5 concentration in the low pollution level is not enough to attract the government’s attention, nor does it cause the health of individual workers to deteriorate. On the contrary, it is likely to represent an increase in the production of firms and even an increase in productivity [29], large-scale production is more conducive to firms to carry out innovative activities and achieve more innovative output. From the perspective of real research, empirical studies by many scholars verified that a certain degree of air pollution promotes the development of firms [40,84], and their conclusions are likely to support this mechanism.

Based on the above speculation, we will further explore and test whether there is an inverted U relationship similar to the Kuznets curve between the PM2.5 and the individual firm innovation output. Similarly, we added the squared term of PM2.5 to the baseline IV regression model for testing. The results in Table 11 show that the quadratic terms of PM2.5 are negatively significant, indicating that there is an inverted U-shaped relationship between PM2.5 and individual firm innovation output. This result verifies our mechanism to some extent: when PM2.5 is at a lower pollution level, PM2.5 has a positive impact on the innovation output of firms, mainly through the mechanism of large-scale development of firms; when the pollution is higher than a certain level, the PM2.5 has a significant negative effect on the innovation of the firm, it mainly acts through the human capital mechanism, and the role of the environmental regulation mechanism is not significant.

### 5.3. Heterogeneity Analysis

We further examine the heterogeneity impact of PM2.5 on firm innovation, and the results are shown in Table 12. First, the sample firms are grouped according to the superior level of PM2.5 concentration (i.e., superior to 35 ug/m^3^). The results show that the PM2.5 has a significant effect on the innovation output of the non-excellent group, but it is not significant for the firms with superior classification. This shows that only when the PM2.5 pollution reaches a certain level (such as when it is at a non-excellent level), its effect on the innovation output of the firm appears, which also verifies the above conclusions about the relationship between PM2.5 and firm innovation.

Secondly, we analyze the impact of PM2.5 on the innovation output of firms with different industrial structure attributes. Regression analysis by grouping according to the level of capital intensity shows that PM2.5 has a significant effect on the innovation output of firms with low capital intensity, but has no obvious effect on firms with high capital intensity. The results of this analysis can also better verify that the mechanism of action of PM2.5 on firm innovation output is mainly human capital mechanism rather than environmental regulation mechanism. On the one hand, industries with high capital intensity are mainly refueling industries and basic industries, including steel smelting, petrochemical, and other industries [85]. Firms in these industries typically have high energy consumption and heavy pollution and are more affected by environmental regulations than other firms. The result of our analysis is that the effect of PM2.5 on such firms is not obvious, which proves that PM2.5 does not have an effect on firm innovation through environmental regulation. On the other hand, industries with low capital intensity mean a larger proportion of labor, and human capital is an important resource for firms to carry out production innovation activities [86]. The effect of PM2.5 on the innovation output of such firms is significant, which means that PM2.5 mainly affects the innovation output of firms by influencing the labor of such firms, and verifies the existence of human capital mechanism.

Finally, we observe the impact of PM2.5 on the innovation output of firms with different productivity levels. Based on the level of labor productivity, we conduct group regression analysis. The results show that PM2.5 has a significant effect on the innovation output of firms with low labor productivity, but not on firms with high labor productivity.

## 6. Conclusions and Discussion

Different from the traditional research on environmental pollution and firm innovation, this paper focuses on the direct influence effect and mechanism of PM2.5 pollution on individual firm innovation output through the identification strategy of constructing instrument variables.

We used the patent data of Chinese listed companies in 2003–2016, and matched the annual PM2.5 concentration data with each firm according to the regional cities, and selected the urban inversion as the instrumental variable of PM2.5 pollution to carry out the regression estimation of the impact effect. The results show that PM2.5 leads to a reduction in the innovation output of firms. Specifically, for every 1 ug/m^3^ increase in the annual average PM2.5 concentration of the city, the total number of patent applications and the total number of invention patent applications of listed companies falls by an average of 7.0%. Based on the existing research conclusions on environmental regulation and firm innovation, we further investigated environmental regulations. Different from the idea that some studies directly internalize environmental regulation into firm specific behaviors (such as pollution investment), we mainly measured environmental regulations in the form of policies, including the “Ecological Demonstration Area”, “Atmospheric Ten Articles”, “Two Control Zones”, and other national-level environmental policies, and the number of environmental regulations in each city. At the same time, we also considered the influence of other pollutants (such as SO_2_ and wastewater) in various cities. The results show that the effect of PM2.5 remains after controlling for environmental regulation. Although the impact of environmental regulation on firm innovation has been controversial in academic circles [43,44,47,87], the empirical results of this paper suggest that the role of PM2.5 affecting firm innovation in environmental regulation is not obvious, and the “Porter hypothesis” of PM2.5 and firm innovation is not established.

After verifying that the environmental regulation mechanism is not established, this paper attempted to explore the mechanism of other PM2.5 affecting the innovation output of firms. Taking into account the direct impact of PM2.5 on individual health, we focused on the possible mechanisms of action from the perspective of human capital. The results show that PM2.5 can significantly worsen the human capital of firms, which may lead to the reduction in innovation output of firms, thus verifying the role of human capital mechanism. In addition, we further tested the relationship between PM2.5 and firm innovation output, and found that there is an inverted U-shaped relationship between PM2.5 and firm innovation output. When PM2.5 is at a lower pollution level, PM2.5 has a positive impact on the innovation output of firms, mainly through the mechanism of large-scale development of firms; when the pollution of PM2.5 is higher than a certain level, PM2.5 has a significant negative effect on firm innovation. It mainly acts through the human capital mechanism, and the role of the environmental regulation mechanism is not significant. The results of heterogeneity analysis can also verify the existence of human capital mechanisms.

The research in this paper has positive guidance and reference significance for understanding the relationship between PM2.5 and firm innovation and formulating corresponding policies. Of course, there are also some shortcomings in this paper. The indicators of innovation output are not rich enough, the description of environmental regulation may be omitted, human capital mechanism testing is not targeted at sample firms, firms’ production and innovation might take place at different hubs, insufficient exogeneity of instrumental variables, measurement errors due to intra-city pollution differences and other issues need to be resolved by future scholars.

## Figures and Tables

**Table 1 ijerph-20-01112-t001:** Descriptive statistics of main variables.

Variables	Obs.	Mean	Std Dev.	Minimum	Maximum
Urban climate data					
PM.5	17,632	39.3487	14.9797	4.5171	90.8565
Thermal inversion	17,632	61.1959	35.0487	1.0000	174.8889
Individual firm data					
Total patent applications	17,632	1.1425	1.4708	0.0000	8.6725
Number of invention patent applications	17,632	0.6963	1.0988	0.0000	8.0665
Utility model patent application amount	17,632	0.7467	1.2021	0.0000	7.6454
Design patent application volume	17,632	0.3040	0.8343	0.0000	6.5265
Total patent authorization	17,632	1.0365	1.3643	0.0000	8.0731
Total number of invention patents	17,632	0.4443	0.8231	0.0000	6.6983
Utility model patent authorization	17,632	0.7257	1.1806	0.0000	7.5909
Design patent authorization	17,632	0.3053	0.8295	0.0000	6.9246
Firm size	17,632	21.6361	1.2346	12.3143	26.8717
Asset–liability ratio	17,632	0.4664	0.2338	0.0512	1.2235
Return on assets	17,632	0.0329	0.0652	−0.2776	0.1970
Capital labor ratio	17,632	12.4195	1.1217	9.3965	15.7480
Firm age	17,632	14.5661	5.5612	0.0000	66.0000
Fixed assets investment	17,632	0.2588	0.1774	0.0022	0.7467
CSS individual data					
Happiness	5937	4.1024	1.1151	1.0000	6.0000
Health satisfaction	5928	6.8679	2.1732	1.0000	10.0000
Pleasure and enjoyment	5937	3.6005	0.8458	1.0000	5.0000
Anger and rage	5937	2.2503	0.7984	1.0000	5.0000
Worry and fear	5937	1.9222	0.8957	1.0000	5.0000
Aversion	5937	1.7189	0.8215	1.0000	5.0000
Medical expenditure	5937	5934.2460	18036.2000	0.0000	500,000.0000
Hourly wage	2840	15.7004	19.3064	0.5556	300.0000
Whether party member	5937	0.1075	0.3097	0.0000	1.0000
Years of education	5937	8.9166	4.2875	0.0000	19.0000
Gender	5937	0.4477	0.4973	0.0000	1.0000
Age	5937	44.3372	12.3387	18.0000	72.0000
Age squared	5937	21.1801	11.0005	3.2400	51.8400
Whether urban registration	5937	0.3396	0.4736	0.0000	1.0000
Whether it works	5937	0.7847	0.4110	0.0000	1.0000
Endowment insurance	5937	0.6311	0.4825	0.0000	1.0000
Medical insurance	5937	0.9052	0.2930	0.0000	1.0000

**Table 2 ijerph-20-01112-t002:** Effects of PM2.5 on innovation.

	FE				IV			
	Total Number of Patent Applications		Total Number of Invention Patent Applications		Total Number of Patent Applications		Total Number of Invention Patent Applications	
	(1)	(2)	(3)	(4)	(5)	(6)	(7)	(8)
PM2.5	0.003	0.003	0.001	0.001	−0.071 ***	−0.070 ***	−0.070 ***	−0.069 ***
	(0.002)	(0.002)	(0.002)	(0.002)	(0.019)	(0.019)	(0.016)	(0.016)
Firm size		0.105 ***		0.092 ***		0.102 ***		0.089 ***
		(0.021)		(0.018)		(0.013)		(0.011)
Asset–liability ratio		0.128 *		0.127 **		0.112 **		0.112 ***
		(0.072)		(0.056)		(0.051)		(0.040)
Return on assets		−0.493 ***		−0.370 ***		−0.572 ***		−0.446 ***
		(0.147)		(0.109)		(0.129)		(0.100)
Capital labor ratio		−0.027		−0.018		−0.029 **		−0.020 *
		(0.018)		(0.016)		(0.012)		(0.010)
Firm age		−0.105 ***		−0.082 ***		−0.430 ***		−0.392 ***
		(0.024)		(0.020)		(0.090)		(0.075)
Investment rate		0.396 ***		0.328 ***		0.399 ***		0.331 ***
		(0.117)		(0.096)		(0.081)		(0.065)
City fixed effects	Y	Y	Y	Y	Y	Y	Y	Y
Year fixed effects	Y	Y	Y	Y	Y	Y	Y	Y
KP LM					103.107 ***	102.821 ***	103.107 ***	102.821 ***
CD Wald					135.505 ***	135.089 ***	135.505 ***	135.089 ***
KP Wald					105.941 ***	105.631 ***	105.941 ***	105.631 ***
N	17,632	17,632	17,632	17,632	17,448	17,448	17,448	17,448
N_g	1967	1967	1967	1967	1783	1783	1783	1783

Notes: *, **, and *** indicate significance at 10%, 5%, and 1% levels, respectively. The standard deviation of robustness is in parentheses. Y indicates that the fixed effect is controlled.

**Table 3 ijerph-20-01112-t003:** Effects of PM2.5 on other types of innovation.

IV	Utility Model Patent Application	Design Patent Application	Utility Model Authorization	Design Authorization	Total Patent Authorization	Total Invention Patent Authorization
	(1)	(2)	(3)	(4)	(5)	(6)
PM2.5	−0.063 ***	−0.026 **	−0.061 ***	−0.037 ***	−0.054 ***	−0.027 **
	(0.016)	(0.011)	(0.017)	(0.011)	(0.018)	(0.012)
Control variable	Y	Y	Y	Y	Y	Y
City fixed effects	Y	Y	Y	Y	Y	Y
Year fixed effects	Y	Y	Y	Y	Y	Y
N	17,448	17,448	17,448	17,448	17,448	17,448
N_g	1783	1783	1783	1783	1783	1783

Notes: **, and *** indicate significance at 5%, and 1% levels, respectively. Control variables include firm size, asset–liability ratio, return on assets, capital labor ratio, firm age, and investment rate. The standard deviation of robustness is in parentheses. Y indicates that the fixed effect or variable has been controlled.

**Table 4 ijerph-20-01112-t004:** Robustness test (1).

IV	Total Number of Patent Applications					Number of Invention Patent Applications				
	(1)	(2)	(3)	(4)	(5)	(6)	(7)	(8)	(9)	(10)
Year fixed effects * Regional fixed effects	−0.057 ***					−0.059 ***				
	(0.017)					(0.014)				
Year fixed effects * Industry fixed effects		−0.087 *					−0.120 ***			
		(0.045)					(0.040)			
Control variable * Time trend			−0.055 ***					−0.057 ***		
			(0.018)					(0.015)		
Control variable * Time trend three terms				−0.054 ***					−0.055 ***	
				(0.018)					(0.015)	
Control variable * Year fixed effects					−0.053 ***					−0.056 ***
					(0.018)					(0.015)
Control variable	Y	Y	Y	Y	Y	Y	Y	Y	Y	Y
City fixed effects	Y	Y	Y	Y	Y	Y	Y	Y	Y	Y
Year fixed effects	Y	Y	Y	Y	Y	Y	Y	Y	Y	Y
N	17,448	17,448	17,448	17,448	17,448	17,448	17,448	17,448	17,448	17,448
N_g	1783	1783	1783	1783	1783	1783	1783	1783	1783	1783

Notes: *, and *** indicate significance at 10%, and 1% levels, respectively. Control variables include firm size, asset–liability ratio, return on assets, capital labor ratio, firm age, and investment rate. The standard deviation of robustness is in parentheses. Y indicates that the fixed effect or variable has been controlled.

**Table 5 ijerph-20-01112-t005:** Robustness test (2).

IV	Total Number of Patent Applications			Number of Invention Patent Applications		
	(1)	(2)	(3)	(4)	(5)	(6)
Delete no patent sample	−0.042 **			−0.036 **		
	(0.018)			(0.016)		
Retain manufacturing samples		−0.067 ***			−0.071 ***	
		(0.024)			(0.021)	
Balance panel			−0.148 **			−0.171 ***
			(0.064)			(0.062)
Control variable	Y	Y	Y	Y	Y	Y
City fixed effects	Y	Y	Y	Y	Y	Y
Year fixed effects	Y	Y	Y	Y	Y	Y
N	8037	10,815	10,304	6450	10,815	10,304
N_g	1246	1169	736	1140	1169	736

Notes: ** and *** indicate significance at 5%, and 1% levels, respectively. Control variables include firm size, asset–liability ratio, return on assets, capital labor ratio, firm age, and investment rate. The standard deviation of robustness is in parentheses. Y indicates that the fixed effect or variable has been controlled.

**Table 6 ijerph-20-01112-t006:** Robustness test (3).

IV	Total Number of Patent Applications			Number of Invention Patent Applications		
	(1)	(2)	(3)	(4)	(5)	(6)
Excluding provincial capitals and above	−0.087 ***			−0.063 ***		
	(0.029)			(0.023)		
Excluding sub-provincial cities		−0.098 ***			−0.093 ***	
		(0.035)			(0.029)	
Control city characteristics			−0.070 ***			−0.067 ***
			(0.021)			(0.017)
Control variable	Y	Y	Y	Y	Y	Y
City fixed effects	Y	Y	Y	Y	Y	Y
Year fixed effects	Y	Y	Y	Y	Y	Y
N	8367	17,448	10,915	8367	17,448	10,915
N_g	866	1783	1102	866	1783	1102

Notes: *** indicate significance at 1% levels, respectively. Control variables include firm size, asset–liability ratio, return on assets, capital labor ratio, firm age, and investment rate. The standard deviation of robustness is in parentheses. Y indicates that the fixed effect or variable has been controlled.

**Table 7 ijerph-20-01112-t007:** Robustness test (4).

IV	The Percentage of Invention Patents	The Percentage of Utility Model Patents	The Percentage of Design Patents
	(1)	(2)	(3)
PM2.5	−0.043 ***	−0.024 **	−0.004 **
	(0.001)	(0.012)	(0.002)
Control variable	Y	Y	Y
City fixed effects	Y	Y	Y
Year fixed effects	Y	Y	Y
N	17,448	17,448	17,448
N_g	1783	1783	1783

Notes: ** and *** indicate significance at 5%, and 1% levels, respectively. Control variables include firm size, asset–liability ratio, return on assets, capital labor ratio, firm age, and investment rate. The standard deviation of robustness is in parentheses. Y indicates that the fixed effect or variable has been controlled.

**Table 8 ijerph-20-01112-t008:** Robustness test (5).

IV	Total Number of Patent Applications				
	(1)	(2)	(3)	(4)	(5)
PM2.5	−0.069 **	−0.069 ***	−0.069 ***	−0.069 **	−0.069 ***
	(0.034)	(0.021)	(0.016)	(0.033)	(0.018)
Control variable	Y	Y	Y	Y	Y
City fixed effects	Y	Y	Y	Y	Y
Year fixed effects	Y	Y	Y	Y	Y
N	17,448	17,448	17,448	17,448	17,448
N_g	1783	1783	1783	1783	1783

Notes: ** and *** indicate significance at 5%, and 1% levels, respectively. Control variables include firm size, asset–liability ratio, return on assets, capital labor ratio, firm age, and investment rate. The standard deviation of cluster robustness is in parentheses, model (1) is clustered to the city level, model (2) clusters to the province level, model (3) clusters to the industry–year level, model (4) clusters to the city–year level, model (5) clusters to the city–industry level. Y indicates that the fixed effect or variable has been controlled.

**Table 9 ijerph-20-01112-t009:** Effect of environmental regulation.

IV	Total Number of Patent Applications					
	Ecological Demonstration Area	Atmospheric Ten Articles	Two Control Zones	Environmental Regulations	Other Pollution	Baseline Regression
	(1)	(2)	(3)	(4)	(5)	(6)
PM2.5	−0.069 ***	−0.086 ***	−0.076 ***	−0.059 ***	−0.067 ***	−0.070 ***
	(0.019)	(0.030)	(0.020)	(0.019)	(0.018)	(0.019)
SO_2_					0.128 ***	
					(0.027)	
Wastewater					−0.056 **	
					(0.023)	
Control variable	Y	Y	Y	Y	Y	Y
City fixed effects	Y	Y	Y	Y	Y	Y
Year fixed effects	Y	Y	Y	Y	Y	Y
N	17,448	17,448	17,448	17,448	17,448	17,448
N_g	1783	1783	1783	1783	1783	1783
**IV**	**Number of Invention Patent Applications**					
	**(7)**	**(8)**	**(9)**	**(10)**	**(11)**	**(12)**
PM2.5	−0.068 ***	−0.077 ***	−0.075 ***	−0.061 ***	−0.066 ***	−0.069 ***
	(0.016)	(0.025)	(0.017)	(0.016)	(0.015)	(0.016)
SO_2_					0.113 ***	
					(0.023)	
Wastewater					−0.035 *	
					(0.019)	
Control variable	Y	Y	Y	Y	Y	Y
City fixed effects	Y	Y	Y	Y	Y	Y
Year fixed effects	Y	Y	Y	Y	Y	Y
N	17,448	17,448	17,448	17,448	17,448	17,448
N_g	1783	1783	1783	1783	1783	1783

Notes: *, **, and *** indicate significance at 10%, 5%, and 1% levels, respectively. Control variables include firm size, asset–liability ratio, return on assets, capital labor ratio, firm age, and investment rate. The standard deviation of robustness is in parentheses. Y indicates that the fixed effect or variable has been controlled.

**Table 10 ijerph-20-01112-t010:** Human capital mechanism test.

	Medical Expenditure	Health Satisfaction	Hourly Wage	Satisfaction of Work	Happiness
	(1)	(2)	(3)	(4)	(5)
pmc2012	791.409 ***	−0.192 ***	−0.706 ***	−0.102 ***	−0.113 ***
	(52.966)	(0.008)	(0.086)	(0.006)	(0.003)
Control variable	Y	Y	Y	Y	Y
City fixed effects	Y	Y	Y	Y	Y
Year fixed effects	Y	Y	Y	Y	Y
N	5937	5928	2840	5929	5937
	Pleasure and enjoyment	Anger and rage	Worry and fear	Aversion	
	(6)	(7)	(8)	(9)	
pmc2012	−0.119 ***	0.020 *	0.011	0.027 ***	
	(0.007)	(0.010)	(0.010)	(0.010)	
Control variable	Y	Y	Y	Y	
City fixed effects	Y	Y	Y	Y	
Year fixed effects	Y	Y	Y	Y	
N	5937	5937	5937	5937	

Notes: *, and *** indicate significance at 10%, and 1% levels, respectively. Control variables include firm size, asset–liability ratio, return on assets, capital labor ratio, firm age, and investment rate. The standard deviation of robustness is in parentheses. Y indicates that the fixed effect or variable has been controlled.

**Table 11 ijerph-20-01112-t011:** Inverted U-shaped relationship of PM2.5 affecting innovation.

IV	Total Number of Patent Applications		Number of Invention Patent Applications	
	(1)	(2)	(3)	(4)
PM2.5	0.293 **	0.165 **	0.133 **	0.123 **
	(0.133)	(0.068)	(0.060)	(0.059)
PM2.5*PM2.5	−0.004 ***	−0.002 ***	−0.002 ***	−0.002 ***
	(0.001)	(0.001)	(0.001)	(0.001)
Control variable	N	Y	N	Y
City fixed effects	N	Y	N	Y
Year fixed effects	N	Y	N	Y
N	17,448	17,448	17,448	17,448
N_g	1783	1783	1783	1783

Notes: **, and *** indicate significance at 5%, and 1% levels, respectively. Control variables include firm size, asset–liability ratio, return on assets, capital labor ratio, firm age, and investment rate. The standard deviation of robustness is in parentheses. N indicates that the fixed effect or variable has not been controlled, Y indicates that the fixed effect or variable has been controlled.

**Table 12 ijerph-20-01112-t012:** Heterogeneity test.

IV	PM2.5 Superior	PM2.5 Non-Superior	High Capital Intensity	Low Capital Intensity	High Labor Productivity	Low Labor Productivity
	(1)	(2)	(3)	(4)	(5)	(6)
PM2.5	−0.090	−0.033 **	0.006	−0.094 ***	−0.028	−0.077 **
	(0.097)	(0.015)	(0.024)	(0.029)	(0.022)	(0.031)
Control variable	Y	Y	Y	Y	Y	Y
City fixed effects	Y	Y	Y	Y	Y	Y
Year fixed effects	Y	Y	Y	Y	Y	Y
N	7583	9673	3980	13,290	5081	12,173
N_g	1036	1217	635	1627	764	1528

Notes: **, and *** indicate significance at 5%, and 1% levels, respectively. Control variables include firm size, asset–liability ratio, return on assets, capital labor ratio, firm age, and investment rate. The standard deviation of robustness is in parentheses. Y indicates that the fixed effect or variable has been controlled.

## Data Availability

The data underlying this article will be shared on reasonable request to the corresponding author (dandanli111@163.com).

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
