# Peer review of "Does PM2.5 (Pollutant) Reduce Firms’ Innovation Output?"

_ijerph, 2023, doi:10.3390/ijerph20021112_

Round 1
Reviewer 1 Report
Review report for ijerph-2119586 Does PM2.5 reduce firms’ innovation output?
Title
PM2.5 is not a common term for all readers. Suggest putting a bracket, what does this PM2.5 refer to? Pollutant?
Abstract
This sentence does not add value to this section. “Scholars tend to pay attention to the impact of air pollution on firms at micro-level.”
What is the research problem?
Introduction
Page 2, row 73. Please elaborate porter hypothesis. “does not explore the direct impact of PM2.5 on individual firm 75 innovation.” This is the research gap highlighted by the authors. What is so special about PM2.5? Please explain.
Page 3, row 123. At the same time, we further control the environmental protection policies 122 such as the “Ecological Demonstration Area”, “Atmospheric Ten Articles”, “Two Control 123 Zones” and the number of regional environmental regulations, as well as other environ-124 mental factors such as pollutant emissions. General readers may not know what are they. The paper does not discuss these issues, just for the sake to examine their impact. It is inappropriate.
The introduction section is too lengthy. Please separate them into the introduction and literature review sections.
variables, data and models
Models should be presented, then followed by variables and data.
Results and analysis
Page 7, row 302. “First, we examine the impact of 302 regional PM2.5 on whether firms choose to innovate, including whether firms apply for 303 patents and whether to apply for invention patents.” How to examine this objective? Furthermore, Table 2 is confusing. Why the dependent variable is “Whether to apply for patent”?
Mechanism testing and heterogeneity analysis
What is the motivation to examine “Human capital mechanism test”? It is not explained under the introduction section.
Too much investigation under this section but they are not being developed well by the research motivation. General readers may not be interested to read the details when the research motivation is not discussed well.
Conclusions
The authors have done a lot of regression tests to examine different impact of independent variables on dependent variables. However, the research motivation is missing. Therefore, this study is less interesting to general readers.
Reviewer 2 Report
1. The introduction section introduces the issue not directly enough, it is suggested that the beginning is closely focused on the importance or shortcomings of enterprise innovation.
2. The introduction is too long, and it is suggested that the literature review in the introduction should be a separate section.
3. It is more common to use data from listed companies as the research sample, and it is suggested to streamline the reasons for the selection.
4. The total number of patents used in the baseline regression is relatively single, and it is suggested to add robustness tests such as patent share.
5. The baseline regression uses ordinary robust standard errors, and it is recommended to replace the clustering robust standard errors in the robustness test section.
6. The conclusion section is a bit lengthy, and it is suggested to further streamline the presentation.
Reviewer 3 Report
The article looks in innovative ways at the impact of pollution on firm innovation and health of the workers. It is an ambitious and rather successful article in my opinion.
English language needs improvement. At least a passage through “grammarly” or other grammar checker will show instances where sentences are not properly constructed. In particular, sentences tend to be too long, with many commas and losing grammatical consistency in the process. One example: “The occurrence of inversion will directly lead to an increase 109 in air pollution, but at the same time, inversion as a naturally occurring meteorological 110 phenomenon, its formation is not affected by regional economic or social factors“. The sentence could be saved changing “as” for “is” and removing the second is: “The occurrence of inversion will directly lead to an increase 109 in air pollution, but at the same time, inversion is a naturally occurring meteorological 110 phenomenon, its formation not affected by regional economic or social factors”. Then starting on its use as IV in a new sentence.
Regarding the identification strategy and the use of thermal inversion, while as the authors argue there might be natural factors leading to thermal inversion irrespective of economic activities, it is difficult to argue that thermal inversion is truly exogenous since it can be affected by policy (eg: the end of smog in London), itself respondent to social and economic conditions. That should be listed as a limitation of the study.
The article is sensitive to the spatial localization of economic activity. While spatial dimension of pollution is clear, the spatial localization of production, innovation and human capital is not so clear. Note also that the are large intra-urban differences in pollution. It seems that measurement is carried out at the city level. This should be mentioned as a limitation. Also, firms production and innovation might take place at different hubs. How is this taken into account? Again, this seems to be a limitation of the data that should be acknowledged.
The article is making statements like “Since PM2.5's impact on firm innovation through environmental regulation is not 430 significant”. Please avoid using significance in this context since it seems to indicate a statistical significance test. It is not the case. You can say that the effect of pollution remains after controlling for regulation or something of the sort.
The article is missing updated references on the topic. Two recent references missing that are relevant for the research and include references to other recent papers:
Wei LY, Liu Z. Air pollution and innovation performance of Chinese cities: human capital and labour cost perspective. Environ Sci Pollut Res Int. 2022 Sep;29(45):67997-68015. doi: 10.1007/s11356-022-20628-w. Epub 2022 May 7. PMID: 35525895.
Lei Z, Huang L, Cai Y. Can environmental tax bring strong porter effect? Evidence from Chinese listed companies. Environ Sci Pollut Res Int. 2022 May;29(21):32246-32260. doi: 10.1007/s11356-021-17119-9. Epub 2022 Jan 11. PMID: 35013959.
Note that the second paper finds strong Porter effects unlike the present one!
Round 2
Reviewer 1 Report
The authors unable to convince the readers why this research is important. Specifically, this sentence is vague. ""Scholars tend to pay attention to the impact of air pollution on firms at micro-level, but little research in the field of firms’ innovation." impact of air pollution on firms? what are the items? Since the research motivation is not strong, it is difficult to convince readers to read this study.
Regarding to this issue in the earlier review report, Modification instruction 7:Regarding how to measure whether a firm chooses to innovate, we use whether a firm applies for a patent as a marker and generate a dummy variable that is set to 1 if the firm applies for a patent and 0 if the firm does not apply for a patent. in addition, Why the dependent variable is "Whether to apply for patent" in Table 2? because, like the total number of patent applications or grants by a firm, whether a firm applies for a patent is also a way for us to measure the innovation of a firm. So we first verified the effect of PM2.5 on whether firms choose to innovate, then we examined the effect of PM2.5 on the number of firm innovations and the number of invention innovations, and finally we examined the effect of PM2.5 on other types of firm innovations. Thanks again!
Where did you clarify this in the revised paper? I would suggest to change the name for dependent variable for Table 2 and 3. Please avoid using "Whether to apply for patent". It is a question, which is inappropriate to label it as dependent variable.
